ecology

biological control, eucalyptus weevil, entomopathogenic nematodes, pathogenicity

**Author for correspondence:**
Bárbara Monteiro de Castro e Castro
e-mail: barbaramcastro@hotmail.com

# *Steinernema diaprepesi* (Rhabditida: Steinernematidae) parasitizing *Gonipterus platensis* (Coleoptera: Curculionidae)

Alixelhe Pacheco Damascena[1], Vanessa Rafaela de Carvalho[1], Murilo Fonseca Ribeiro[1], André Ballerini Horta[1], Bárbara Monteiro de Castro e Castro[2], Antônio José Vinha Zanuncio[3], Carlos Frederico Wilcken[1], José Cola Zanuncio[2] and Silvia Renata Siciliano Wilcken[1]

[1]Universidade Estadual Paulista (UNESP), Faculdade de Ciências Agronômicas, Departamento de Proteção Vegetal, Departamento de Proteção Vegetal, 18610-034, Botucatu, São Paulo, Brazil
[2]Departamento de Entomologia/BIOAGRO, and [3]Departamento de Engenharia Florestal, Universidade Federal de Viçosa, Viçosa, Minas Gerais, 36570-900 Brazil

APD, 0000-0003-1374-5119; VRC, 0000-0002-2229-464X;
MFR, 0000-0003-1909-8709; ABH, 0000-0002-2692-9024;
BMCC, 0000-0002-7965-0270; AJVZ, 0000-0001-5145-4027;
CFW, 0000-0001-9875-4158; JCZ, 0000-0003-2026-281X;
SRSW, 0000-0002-9306-0197

Entomopathogenic nematodes (EPNs) can control pests due to mutualistic association with bacteria that reproduce and kill the host from septicemia, making the environment favourable for nematode development and reproduction. The objective of this study was to identify an EPN isolate collected in eucalyptus cultivation and to determine its pathogenicity with regard to *Gonipterus platensis* Marelli (Coleoptera: Curculionidae). Four steel-mesh traps with two seventh-instar *Galleria mellonella* larvae were buried 5 cm deep in the soil in a commercial *Eucalyptus* plantation. After 7 days, the traps were packed in plastic bags and transported to laboratory to isolate the EPNs using White traps. The obtained nematodes were multiplied in *G. mellonella* larvae and identified by sequencing their D2/D3 expansion of the 28S rDNA region by polymerase chain reaction (PCR) and specific primers for ITS regions.

*Steinernema diaprepesi* was identified and inoculated into *G. platensis* pupae at doses of 500, 1000 and 5000 infective juveniles (IJs) to determine its pathogenicity to this pest. At 8 days after inoculation, the mortality rate of the *G. platensis* pupae was 80% with the lowest concentration and 100% with the others. The emergence of nematodes and the rapid degradation of *G. platensis* pupae were observed in those inoculated with IJs. The pathogenicity to the *G. platensis* pupae indicates potential for using this nematode in the integrated management of this insect.

# 1. Introduction

*Gonipterus platensis* Marelli (Coleoptera: Curculionidae), the eucalyptus snout beetle, native to Tasmania, is one of the major *Eucalyptus* pests. This beetle was reported in New Zealand in 1890 and causes crop damage in South America, southwestern North America (California), the Iberian Peninsula, the Canary Islands and Hawaii [1]. The attack severity and high dispersal capacity justify efforts to manage this pest [2]. Adults and, mainly, *G. platensis* larvae which feed exclusively on young eucalyptus leaves can damage this plant. Severe attacks and successive defoliation can cause shoot death, reduced growth, bifurcation and warping of the plant stem [3].

Biological control with the *Anaphes nitens* Girault (Hymenoptera: Mymaridae) egg parasitoid, native to Australia, is the main management strategy for the eucalyptus weevil [2]. The control efficiency of this pest is low in many regions, due to factors such as altitude and parasitoid–host incompatibility [1,4]. However, losses caused by the eucalyptus weevil are two to three times lower with biological control than with other management strategies, such as replacement of genetic material and use of insecticides [5].

The identification of natural enemies acting together with *A. nitens* can increase the control efficiency of *Gonipterus* spp., as reported with the parasitoid *Anaphes inexpectatus* [6]. Although a single natural enemy may adequately suppress an invasive pest, multi-species introductions are a frequent practice in classical biological control, increasing the mortality of pests [7]. For this reason, new biological control agents that might complement the activity of *A. nitens* have to be identified. The search for natural enemies for this pest has been focused on eggs, larvae and adults of this pest, present in the upper third of the host trees [8–10], as reported with parasitoids *A. tasmaniae*, *A. inexpectatus*, *Cirrospilus* sp. and *Euderus* sp. [11], while control of the pupae of this pest, which occurs in the soil, has been deficient.

Entomopathogenic nematodes (EPNs) (Steinernematidae and Heterorhabditidae) are effective in pest management due to their association with bacteria of the genus *Xenorhabdus* (Thomas, Poinar) and *Photorhabdus* (Boemare; Louis, Kuhl) [12]. Infective juveniles (IJs) of these nematodes release the bacteria into the haemocele, killing the host by septicemia in 24 to 48 h, and making the environment favourable for its development and reproduction [13].

Population surveys of EPNs that inhabit the soil of different environments [14,15] are important for identifying their species and efficacy in biological control programmes [16]. The objective of this study was to identify EPN species in eucalyptus cultivation and to evaluate their pathogenicity for *G. platensis*.

# 2. Material and methods

## 2.1. Nematode survey

Four steel-mesh traps with two seventh instars of *Galleria mellonella* L. (Lepidoptera: Pyralidae) larvae each were buried in the soil at a depth of 5 cm in a commercial *Eucalyptus grandis × Eucalyptus urophylla* plantation (22°59′49″ S, 48°29′57″ W, 870 m) where *G. platensis* larvae and adults were present. After 7 days, the traps were removed from the soil, packed in plastic bags and transported to the Nematology Laboratory of the Faculdade de Ciências Agronômicas of the Universidade Estadual Paulista (UNESP) for isolation and identification of entomopathogenic nematodes.

*Galleria mellonella* larvae were removed from the traps, washed in sodium hypochlorite solution (1%) and transferred to White traps [17] in an incubator (BOD) at 25°C for 21 days. Infective juveniles (IJs) obtained from these traps were inoculated in *G. mellonella* larvae to identify the nematode.

## 2.2. Multiplication of the isolate

Infectious juveniles (IJs) were multiplied in *G. mellonella* larvae. Five insect larvae were placed per Petri dish (9 cm in diameter × 1.5 cm in height), which were coated with filter paper moistened with a nematode suspension at a concentration of 500 IU cm$^{-2}$. Dead larvae were transferred to White traps

**Table 1.** Primers used to amplify the genomic DNA of entomopathogenic nematodes obtained from soil in an area with eucalyptus plantation.

| primers | direction | sequence |
| --- | --- | --- |
| D2A[a] | forward | CAAGTACCGTGAGGGAAAGTTG |
| D3B[a] | reverse | TCGGAAGGAACCAG CTACTA |
| KN58[b] | forward | GTATGTTTGGTTGAAGGTC |
| KNRV[b] | reverse | CACGCTCATACAACTGCTC |
| DIAPR1A | forward | CGTAGGTGAACCTGCGGAAG |
| DIAPR1B | reverse | GTTCAGCGGGTAGTCTTGCT |
| DIAPR2A | forward | ACTGCTTCTCTGAGCGCTTT |
| DIAPR2B | reverse | CCTCCATTAGCCCATCGCAT |

[a]Al-Banna et al. [20].
[b]Nguyen et al. [21].

[17] and stored in an incubator (BOD) at 25°C for 21 days. The IJs obtained from these hosts were collected in water film (1 cm deep) in Erlemeyers kept in an incubation chamber (BOD) at a temperature of 18°C and 70% RH.

## 2.3. Molecular identification

The genomic DNA of 50 IJs, isolated from the samples, was extracted in 50 µl of 0.85% NaCl with the modified worm lysis buffer (WLB) extraction method [18,19]. DNA samples from EPNs were plated with KCl (50 mM), Tris (10 mM pH 8.2), MgCl 2 (2.5 mM), Tween 20 (0.45%) and proteinase K (20 mg ml$^{-1}$) without gelatin (0.05%). These samples were left at −70°C for 15 min, incubated for one hour at 60°C, then 15 min at 95°C and stored at −20°C.

The extracted DNA was amplified with the polymerase chain reaction (PCR) technique with the universal primers D2A and D3B for the expansion of 28S rDNA sequence [20], internal to the ITS, KN58 and KNRV [21] and the specific drawn (table 1). The reactions were performed in an Infinigen thermal cycler (model TC-96CG) in 0.5 ml tubes with 12.5 µl of Polymerase Mix Master Red (Neobio), 7.5 µl of nuclease-free water (Promega), 1 µl of each primer (10 mM) and 3 µl of genomic DNA per sample, totalling 25 µl of solution per tube and reaction. The cycles for the universal primers D2A and D3B were preceded by initial denaturation at 94°C for 7 min, followed by 35 cycles of denaturation at 94°C for 1 min, annealing at 55°C for 1 min, extension at 72°C for 1 min and final extension of 72°C for 10 min [22]. The primers KN58 and KNRV were amplified by PCR according to the established cycling [21], and those drawn were preceded by initial denaturation at 94°C for 7 min, followed by 35 denaturation cycles at 94°C for 1 min, annealing at 58°C for 1 min, extension at 72°C for 1 min and final extension of 72°C for 10 min. Negative controls with 3 µl of water were added in the assays to check for possible PCR reagent contaminations. PCR amplification products were visualized by 1% agarose gel electrophoresis with 100 bp marker (Norgen) and UV light Transilluminator (Major Science).

The PCR product was purified according to recommendations for the Celcco PCR purification kit (Qiagen, Cat # 14400) and sequenced in a Sanger automated DNA sequencer (model ABI 3500, Applied Biosystems) at the Institute of Biotechnology (IBTEC) of the Universidade Estadual Paulista (UNESP). The species was identified by the sequences obtained, which were aligned and compared in the BLAST with the data deposited in GenBank (http://www.ncbi.nlm.nih.gov). Specific primers, based on the genomic sequence obtained, were designed after sequencing and tested in populations of *Steinernema feltiae* (Filipjev) and *S. glaseri* (Steiner), phylogenetically close to *S. diaprepesi* [23], to prove their specificity. These primers were annealed at 50–60°C with 40–60% guanine and cytosine without formation of secondary structures, region 18 to 22 nucleotides and PCR products between 500 and 1200 bp [24,25].

## 2.4. Obtaining *Gonipterus platensis*

*Gonipterus platensis* pupae were obtained from artificial rearing of the Laboratory of Biological Control of Forest Pests (LCBPF), UNESP, Campus of Botucatu. This insect was reared in an air-conditioned room at

**Table 2.** Mortality of *Gonipterus platensis* (Coleoptera: Curculionidae) and number of individuals (mean ± s.e.) of *Steinernema diaprepesi* (Rhabditida: Steinernematidae) emerged per pupa of this insect 8 days after inoculation with different concentrations of infective juveniles (IJs). Means followed by the same letter, per column, do not differ at 5% probability level by the Tukey test ($p < 0.05$).

| treatments | mortality (%) | emergence |
| --- | --- | --- |
| control | 0 ± 0 a | 0 ± 0 a |
| 500 IJs | 80 ± 13.3 b | 33735 ± 7913 b |
| 1000 IJs | 100 ± 0.0 b | 35625 ± 6556 b |
| 5000 IJs | 100 ± 0.0 b | 32920 ± 6523 b |
| CV (%) | 31.51 | 51.72 |

$25 \pm 1°C$, relative humidity of $50 \pm 10\%$ and photoperiod 12 : 12 h (L : D). Adults of these insects were kept in wooden cages ($40 \times 45 \times 80$ cm) covered with voile and fed with *Eucalyptus urophylla* leaves, where they oviposited. The *G. platensis* larvae were fed with tender *E. urophylla* leaves in branch bundles.

## 2.5. Pathogenicity of the entomopathogenic nematode to *Gonipterus platensis*

*Gonipterus platensis* pre-pupae were individualized in 50 ml plastic pots with 32 g of autoclaved and sieved sand and 3 ml of distilled water, mixed with the aid of a sterile rod. After 15 days, 2 ml of the suspension with 500, 1000 or 5000 IU of 48 h old EPNs were added to the substrate. Pupae in the control received 2 ml of distilled water.

The plastic pots with *G. platensis* pupae were kept in an incubator (BOD) at $25 \pm 2°C$ and $70 \pm 10\%$ RH. Mortality of these pupae was evaluated 8 days after inoculation of the nematode. Dead pupae were individualized in White traps [17] for the emergence of the nematodes and number of individuals of *S. diaprepesi* emerged per pupa of this insect was counted.

The experiment was conducted according to a completely randomized design, with four treatments and 10 replications, each with one with *G. platensis* pre-pupae. The data were analysed using the statistical software SISVAR 5.6 and the means compared by the Tukey test at 5% significance [26].

# 3. Results

## 3.1. Molecular identification of the nematode

*Steinernema diaprepesi* (Rhabditida: Steinernematidae) was identified with 100% similarity with the nucleotide sequences obtained from the Sanger (accession number GU173994.1).

The primers KN58 and KNRV of the ITS region did not amplify DNA fragments of *S. diaprepesi*. Pairs of specific primers designed with DIAPR1A/DIAPR1B and DIAPR2A/DIAPR2B amplified products of 830 and 682 bp, respectively, with specificity for *S. diaprepesi*. Sequences obtained from PCR products showed 100% similarity to *S. diaprepesi* (accession number GU173996.1) when submitted to sequencing and BLAST. The drawn primers did not amplify PCR products from *S. feltiae* and *S. glaseri* populations phylogenetically close to *S. diaprepesi*.

## 3.2. Pathogenicity of *Steinernema diaprepesi* to *Gonipterus platensis*

*Steinernema diaprepesi* infected, reproduced and killed the *G. platensis* pupae 8 days after its inoculation. Pupae parasitized had a darker colour with orange or brown tones and no movement. The three IJ concentrations caused mortality of this host with high emergence of nematodes ($p < 0.05$) that quickly disintegrated the host pupae. The mortality of this host at the highest (1000 and 5000) and intermediate (500) concentrations of IJs was 100 and 80%, respectively. No pupae of *G. platensis* died in the control without emergence of nematodes (table 2).

# 4. Discussion

The identification of *S. diaprepesi* from molecular analyses with high similarity has also been performed in Argentina [27] and Mexico [28], and for other nematodes as *Heterorhabditis amazonensis* (Andaló, 2006), *Metarhabditis rainai* (Carta & Osbrink, 2005), *Oscheius tipulae* (Lam; Webster, 1971) and *S. rarum* (De Doucet, 1986) in Brazil [16] using Sanger sequencing. This nematode *S. diaprepesi* was first identified in *Diaprepes abbreviates* Linnaeus (Coleoptera: Curculionidae) larvae in Florida, USA [23] and, in South America, only in Venezuela and Argentina [27]. *Steinernema diaprepesi* is highly virulent for lepidopteran larvae such as those of *Galleria mellonella* L. (Lepidoptera: Pyralidae) and *Spodoptera frugiperda* JE Smith (Lepidoptera: Noctuidae), due to its association with the symbiotic bacterium *Xenorhabdus doucetiae* Tailliez, Pagès, Ginibra & Boemare, 2006 [27,29]. Therefore, the detection of *S. diaprepesi* in eucalyptus crops in Brazil can contribute to programmes for integrated management in this culture in the country.

The DNA amplification absence in the primers for *S. feltiae* and *S. glaseri* shows specificity to the *S. diaprepesi* [23]. Identification with specific primers to species which are morphologically similar allows obtaining accurate, reliable and useful results for processing a large number of specimens in a single assay and to identifying the desired species at a reduced cost [30].

The high death of *G. platensis* pupae after inoculation of the three IJ concentrations confirms their susceptibility to *S. diaprepesi* and the potential of this nematode for managing this pest, as reported for *G. mellonella* and *S. frugiperda* [29]. The *G. platensis* mortality rate is related to the association of *S. diaprepesi* with the symbiotic bacterium *X. doucetiae* [27], which when released into host haemocele, promoted rapid degradation of the host pupae. The high number of juveniles multiplied in the corpse of the host, evidenced by the number of individuals emerged, accelerates its degradation due to competition for food and space, depleting the corpse nutrients, which can reduce the number of nematode generations and consequently the number of emerged IJs [31].

# 5. Conclusion

The collection of *S. diaprepesi* in an area with *G. platensis* and the high pathogenicity of this nematode indicate its potential for the integrated management of this pest. This is important because native EPN species exclude risks associated with the introduction of exotic species.

Data accessibility. The data from the survey is freely available on GenBank at https://www.ncbi.nlm.nih.gov/nuccore/ MT121972 [32].

Authors' contributions. A.P.D., V.R.C., M.F.R., A.B.H., C.F.W. and S.R.S.W. designed and performed the experiments and analysed the data, and A.J.V.Z., A.P.D., B.M.C.C and J.C.Z. wrote and edited the manuscript. All authors read and approved the final manuscript.

Competing interests. We have no competing interests.

Funding. We thank the Brazilian institutions 'Conselho Nacional de Desenvolvimento Científico e Tecnológico (CNPq)', 'Coordenação de Aperfeiçoamento de Pessoal de Nível Superior- Brasil (CAPES)', 'Fundação de Amparo à Pesquisa do Estado de Minas Gerais (FAPEMIG)', 'Programa Cooperativo sobre Proteção Florestal/PROTEF do Instituto de Pesquisas e Estudos Florestais/IPEF' and 'KOPPERT' for financial support.

Acknowledgements. David Michael Miller, a professional editor and proofreader and native English speaking, has reviewed and edited this article for structure, grammar, punctuation, spelling, word choice and readability.

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
