## [Reviewer comments · Royal Society Open Science]

Review History

RSOS-200282.R0 (Original submission)

Review form: Reviewer 1 (Marcus Soares)

Is the manuscript scientifically sound in its present form?

Yes

Are the interpretations and conclusions justified by the results?

Yes

Is the language acceptable?

Yes

Do you have any ethical concerns with this paper?

No

Have you any concerns about statistical analyses in this paper?

No

Recommendation?

Accept with minor revision (please list in comments)

Comments to the Author(s)

The manuscript is interesting, well written and aims identify an EPN isolate collected in eucalyptus cultivation and to determine its pathogenicity with regard to *Gonipterus platensis*. This insect is an important pest worldwide and the identification of the pathogen may increase the IPM tools for this pest. The information can also be useful for the development of new bio-based products. I recommend the publication with minor corrections, according to the attached file (Appendix A).

Review form: Reviewer 2**Is the manuscript scientifically sound in its present form?**

No

Are the interpretations and conclusions justified by the results?

No

Is the language acceptable?

Yes

Do you have any ethical concerns with this paper?

No

Have you any concerns about statistical analyses in this paper?

No

Recommendation?

Reject

Comments to the Author(s)

In the attached file (Appendix B) you can see the comments.

Decision letter (RSOS-200282.R0)

Dear Dr Castro,

The editors assigned to your paper ("*Steinernema diaprepesi* (Rhabditida: Steinernematidae) parasitizing *Gonipterus platensis* (Coleoptera: Curculionidae)") have now received comments from reviewers. We would like you to revise your paper in accordance with the referee and Associate Editor suggestions which can be found below (not including confidential reports to the Editor). Please note this decision does not guarantee eventual acceptance.

Please submit a copy of your revised paper before 04-Jun-2020. Please note that the revision deadline will expire at 00.00am on this date. If we do not hear from you within this time then it

will be assumed that the paper has been withdrawn. In exceptional circumstances, extensions may be possible if agreed with the Editorial Office in advance. We do not allow multiple rounds of revision so we urge you to make every effort to fully address all of the comments at this stage. If deemed necessary by the Editors, your manuscript will be sent back to one or more of the original reviewers for assessment. If the original reviewers are not available, we may invite new reviewers.

- Data accessibility

If you wish to submit your supporting data or code to Dryad (<http://datadryad.org/>), or modify your current submission to dryad, please use the following link:
<http://datadryad.org/submit?journalID=RSOS&manu=RSOS-200282>

- Competing interests

- Authors' contributions

- Acknowledgements

- Funding statement

Kind regards,

Andrew Dunn

on behalf of Dr Richard Benton (Associate Editor) and Kevin Padian (Subject Editor)

Editor comments:

One reviewer is quite favorable and offers useful comments. The other is less so but it seems that these could be relatively easily addressed because they don't question the main science of the paper. Please address all comments and best wishes for revising.

Comments to Author:

Reviewers' Comments to Author:

Reviewer: 1

Comments to the Author(s)

The manuscript is interesting, well written and aims identify an EPN isolate collected in eucalyptus cultivation and to determine its pathogenicity with regard to *Gonipterus platensis*. This insect is an important pest worldwide and the identification of the pathogen may increase the IPM tools for this pest. The information can also be useful for the development of new bio-based products. I recommend the publication with minor corrections, according to the attached file.

Reviewer: 2

Comments to the Author(s)

In the attached file you can see the comments.

Author's Response to Decision Letter for (RSOS-200282.R0)

See Appendix C.

RSOS-200282.R1 (Revision)

Review form: Reviewer 2

Is the manuscript scientifically sound in its present form?

Yes

Are the interpretations and conclusions justified by the results?

Yes

Is the language acceptable?

Yes

Do you have any ethical concerns with this paper?

No

Have you any concerns about statistical analyses in this paper?

No

Recommendation?

Accept as is

Comments to the Author(s)

No comments.

Decision letter (RSOS-200282.R1)

Dear Dr Castro,

It is a pleasure to accept your manuscript entitled "Steinernema diaprepesi (Rhabditida: Steinernematidae) parasitizing *Gonipterus platensis* (Coleoptera: Curculionidae)" in its current form for publication in Royal Society Open Science. The comments of the reviewer(s) who reviewed your manuscript are included at the foot of this letter.

on behalf of Dr Richard Benton (Associate Editor) and Kevin Padian (Subject Editor)
openscience@royalsociety.org

Reviewer comments to Author:

Reviewer: 2

Comments to the Author(s)

No comments.

Appendix A**ROYAL SOCIETY
OPEN SCIENCE****Steinernema diaprepesi (Rhabditida: Steinernematidae)
parasitizing Gonipterus platensis (Coleoptera:
Curculionidae)**

Journal:	Royal Society Open Science
Manuscript ID	RSOS-200282
Article Type:	Research
Date Submitted by the Author:	06-Mar-2020
Complete List of Authors:	Damascena, Aixelhe; Universidade Estadual Paulista, Departamento de Proteção Vegetal Carvalho, Vanessa; Universidade Estadual Paulista, Departamento de Proteção Vegetal Ribeiro, Murilo; Universidade Estadual Paulista, Departamento de Proteção Vegetal Horta, André; Universidade Estadual Paulista, Departamento de Proteção Vegetal Castro, Bárbara; Universidade Federal de Viçosa, Wilcken, Carlos; Universidade Estadual Paulista, Departamento de Proteção Vegetal Zanuncio, José; Universidade Federal de Viçosa, Departamento de Biologia Animal; Wilcken, Sílvia; Universidade Estadual Paulista, Departamento de Proteção Vegetal
Subject:	ecology < BIOLOGY
Keywords:	Biological control, eucalyptus weevil, entomopathogenic nematodes, pathogenicity
Subject Category:	Organismal and Evolutionary Biology

Author-supplied statements

Relevant information will appear here if provided.

Ethics

Does your article include research that required ethical approval or permits?:

This article does not present research with ethical considerations

Statement (if applicable):

CUST_IF_YES_ETHICS :No data available.

Data

It is a condition of publication that data, code and materials supporting your paper are made publicly available. Does your paper present new data?:

Yes

Statement (if applicable):

The data from the survey is freely available on GenBank at <https://www.ncbi.nlm.nih.gov/nuccore/MT121972> (Damascena et al., 2020)

Conflict of interest

I/We declare we have no competing interests

Statement (if applicable):

CUST_STATE_CONFLICT :No data available.

Authors' contributions

This paper has multiple authors and our individual contributions were as below

Statement (if applicable):

A.P.D., V.R.C., M.F.R., A.B.H., C.F.W. and S.R.S.W. performed experiments, analyzed the data and designed experiments, A.P.D., B.M.C.C and J.C.Z. wrote and edited the manuscript. All authors read and approved the final manuscript.

***Steinernema diaprepesi* (Rhabditida: Steinernematidae) parasitizing *Gonipterus platensis* (Coleoptera: Curculionidae)**

Alixelhe Pacheco Damascena¹; Vanessa Rafaela de Carvalho¹, Murilo Fonseca Ribeiro¹, André Ballerini Horta¹, Bárbara Monteiro de Castro e Castro², Carlos Frederico Wilcken¹, José Cola Zanuncio², Silvia Renata Siciliano Wilcken¹

¹ Faculdade de Ciências Agronômicas, Departamento de Proteção Vegetal, Universidade Estadual Paulista (UNESP), 18610-034, Botucatu, São Paulo, Brasil.

² Departamento de Entomologia/BIOAGRO, Universidade Federal de Viçosa, 36570-900, Viçosa, Minas Gerais, Brasil.

Keywords: Biological control, eucalyptus weevil, entomopathogenic nematodes, pathogenicity.

1. Summary

Entomopathogenic nematodes (EPNs) can control pests due to the mutualistic association with bacteria. The objective of this study was to identify an EPN isolate collected in eucalyptus cultivation and to determine its pathogenicity with regard to *Gonipterus platensis*. Four steel-mesh traps with two seventh-instar *Galleria mellonella* larvae were buried 5 cm deep in the soil in a commercial *Eucalyptus* plantation. After seven days, the traps were packed in plastic bags and transported to laboratory to isolate the EPNs using White traps. The obtained nematodes were multiplied in *G. mellonella* larvae and identified by sequencing their D2/D3 expansion of the 28S rDNA region by PCR and specific primers for ITS regions. *Steinernema diaprepesi* was identified and inoculated into *G. platensis* pupae at doses of 500, 1,000 and 5,000 infective juveniles (IJs) to determine its pathogenicity to this pest. At eight days after inoculation, the mortality rate of the *G. platensis* pupae was 80% with the lowest concentration and 100% with the others. The emergence of nematodes and the rapid degradation of *G. platensis* pupae were observed in those inoculated with IJs. The pathogenicity to the *G. platensis* pupae indicates potential for using this nematode in the integrated management of this insect.

2. Introduction

Gonipterus platensis Marelli (Coleoptera: Curculionidae), the eucalyptus weevil, native to Tasmania, is one of the major *Eucalyptus* pests. This beetle was reported in New Zealand in 1890 and causes crop damage in South America, southwestern North America (California), the Iberian Peninsula, the Canary Islands and Hawaii (Mapondera et al., 2012). The attack severity and high dispersal capacity justify efforts to manage this pest (Jegger et al., 2018). Adults and, mainly, *G. platensis* larvae which feed exclusively on young eucalyptus leaves can damage this plant. Severe attacks and successive defoliation can cause shoot death, reduced growth, bifurcation and warping of the plant stem (Souza et al., 2016).

Biological control with the *Anaphes nitens* Girault (Hymenoptera: Mymaridae) egg parasitoid, native to Australia, is the main management strategy for the eucalyptus weevil (Jegger et al., 2018). The control efficiency of this pest is low in many regions, due to factors such as altitude and parasitoid-host

*Author for correspondence (barbaramcastro@hotmail.com).

†Present address: Departamento de Entomologia/BIOAGRO, Universidade Federal de Viçosa, Viçosa, Minas Gerais, 36570-900, Brasil.

incompatibility (Mapondera et al., 2012; Reis et al., 2012). However, losses caused by the eucalyptus weevil are two to three times lower with biological control than with other management strategies, such as replacement of genetic material and use of insecticides (Valente et al., 2018).

The identification of natural enemies acting together with *A. nitens*, can increase the control efficiency of *Gonipterus* spp. In general, this search has been focused on natural enemies of the egg, larva and adult stages of this pest, present in the upper third of the trees (Gumovsky et al., 2015; Nascimento et al., 2017; Valente et al., 2017), while control of the pupae of this pest, which occurs in the soil, has been deficient.

Entomopathogenic nematodes (EPNs) (Steinernematidae and Heterorhabditidae) are effective in pest management due to their association with bacteria of the genus *Xenorhabdus* (Thomas, Poinar) and *Photorhabdus* (Boemare; Louis, Kuhl) (Batalla-Carrera et al., 2016.). Infective juveniles (IJs) of these nematodes release the bacteria into the hemocele, killing the host by septicemia in 24 to 48 hours, and making the environment favorable for its development and reproduction (Poinar, 1990).

Population surveys of entomopathogenic nematodes (EPNs), that inhabit the soil of different environments (Tarasco et al., 2015; Tumialis et al., 2016), are important for identifying their species and efficacy in biological control programs (De Brida et al., 2017).

The objective of this study was to identify EPN species in eucalyptus cultivation and to evaluate their pathogenicity for *G. platensis*.

3. Materials and Methods

Nematode Survey

Four steel-mesh traps with two seventh instars of *Galleria mellonella* L. (Lepidoptera: Pyralidae) larvae each were buried in the soil at a depth of 5 cm in a commercial *Eucalyptus grandis* × *Eucalyptus urophylla* plantation where *G. platensis* larvae and adults were present. After seven days, the traps were removed from the soil, packed in plastic bags and transported to the FCA/UNESP Nematology Laboratory for isolation and identification of entomopathogenic nematodes.

Galleria mellonella larvae were removed from the traps, washed in sodium hypochlorite solution (1%) and transferred to White traps (White, 1927) in an incubator (B.O.D.) at 25 °C for 21 days. Infective juveniles (IJs) obtained from these traps were inoculated in *G. mellonella* larvae to identify the nematode.

Multiplication of the Isolate

Infectious juveniles (IJs) were multiplied in *G. mellonella* larvae. Five insect larvae were placed per Petri dish (9 cm in diameter × 1.5 cm in height), which were coated with filter paper moistened with a nematode suspension at a concentration of 500 IU/cm². Dead larvae were transferred to White traps (White, 1927) and stored in an incubator (B.O.D.) at 25 °C for 21 days. The IJs obtained from these hosts were collected in water film (1 cm deep) in Erlemeyers kept in an incubation chamber (B.O.D.) at a temperature of 18 °C and 70% RH.

Molecular Identification

The genomic DNA of 50 IJs, isolated from the samples, was extracted in 50 µl of 0.85% NaCl with the modified WormLysis Buffer (WLB) extraction method (Carvalho et al., 2018; Williams et al., 1992). DNA samples from EPNs were plated with KCl (50 mM), Tris (10 mM pH 8.2), MgCl₂ (2.5 mM), Tween 20 (0.45%) and proteinase K (20 mg/ml) without gelatin (0.05%). These samples were left at -70 °C for 15 minutes, incubated for one hour at 60 °C, then 15 minutes at 95 °C and stored at -20 °C.

The extracted DNA was amplified with the Polymerase Chain Reaction (PCR) technique with the universal primers D2A and D3B for the expansion of 28S rDNA sequence (Al-Banna et al., 2004), internal to the ITS, KN58 and KNRV (Nguyen et al., 2001) and the specific drawn (Table 1). The reactions were performed in an INFINIGEN Thermal Cycler (model TC-96CG) in 0.5 ml tubes with 12.5 µl of Polymerase Mix Master Red (Neobio), 7.5 µl of NucleaseFreeWater (Promega), 1 µl of each primer (10 mM) and 3 µl of genomic DNA per sample, totaling 25 µl of solution per tube and reaction. The cycles for the universal primers D2A and D3B were preceded by initial denaturation at 94 °C for seven minutes, followed by 35 cycles of denaturation at 94 °C for one minute, annealing at 55 °C for one minute, extension at 72 °C for one minute and final extension of 72 °C for ten minutes (Mracek et al., 2006). The primers KN58 and KNRV were amplified by PCR according to the established cycling (Nguyen et al., 2001), and those drawn were preceded by initial denaturation at 94 °C for seven minutes, followed by 35 denaturation cycles at 94 °C for one minute,

annealing at 58 °C for one minute, extension at 72 °C for one minute and final extension of 72 °C for ten minutes. Negative controls with 3µl of water were added in the assays to check for possible PCR reagent contaminations. PCR amplification products were visualized by 1% agarose gel electrophoresis with 100 bp marker (Norgen) and UV Light Transilluminator (Major Science).

The PCR product was purified according to recommendations for the Qiagen PCR Purification Kit (Qiagen, Cat # 14400) and sequenced in a Sanger Automated DNA Sequencer (Model: ABI 3500- Applied Biosystems) at the Institute of Biotechnology (IBTEC-UNESP). The species was identified by the sequences obtained, which were aligned and compared in the BLAST with the data deposited in GenBank (<http://www.ncbi.nlm.nih.gov>). Specific primers, based on the genomic sequence obtained, were designed after sequencing and tested in populations of *Steinernema feltiae* (Filipjev) and *Steinernema glaseri* (Steiner), phylogenetically close to *S. diaprepesi* (Nguyen & Duncan, 2002), to prove their specificity. These primers were annealed at 50-60 °C with 40% to 60% guanine and cytosine without formation of secondary structures, region 18 to 22 nucleotides and PCR products between 500 and 1200 bp (Freeland, 2016, Abd-Elsalam et al., 2003).

Obtaining *Gonipterus platensis*

Gonipterus platensis pupae were obtained from the Laboratory of Biological Control of Forest Pests (LCBPF) of the Faculdade de Agronomia of the Universidade Estadual Paulista (FCA/UNESP). This insect was created in an air-conditioned room at 25 ± 1 °C, relative humidity of 50 ± 10% and photoperiod 12: 12h (L: D). Adults of these insects were kept in wooden cages (40 x 45 x 80 cm) covered with *voil* and fed with *Eucalyptus urophylla* leaves, where they oviposited. The *G. platensis* larvae were fed with tender *Eucalyptus urophylla* leaves in branch bundles.

Pathogenicity of the Entomopathogenic Nematode to *G. platensis*

Gonipterus platensis pre-pupae were individualized in 50 ml plastic pots with 32 g of autoclaved and sieved sand and 3 ml of distilled water, mixed with the aid of a sterile rod. After 15 days, 2 ml of the suspension with 500, 1000 or 5000 IU of 48-hour-old EPNs were added to the substrate. Pupae in the control received 2 mL of distilled water.

The plastic pots with *G. platensis* pupae were kept in an incubator (B.O.D.) at 25 ± 2 °C and 70 ± 10% RH. Mortality of these pupae was evaluated eight days after inoculation of the nematode. Dead pupae were individualized in White traps (White, 1927) for the emergence of the nematodes.

The experiment was conducted according to a completely randomized design, with four treatments and 10 replications, with one with *G. platensis* pre-pupae. The data were analyzed using the statistical software SISVAR 5.6 and the means compared by the Tukey test at 5% significance (Ferreira, 2015).

4. Results

Molecular Identification of the Nematoid

The nucleotide sequences obtained from the Sanger sequencing showed 100% similarity to those of *Steinernema diaprepesi* (Rhabditida: Steinernematidae) (accession number GU173994.1).

Steinernema diaprepesi was identified with primers KN58 and KNRV of the ITS region, which did not amplify DNA fragments of this nematode. Pairs of specific primers designed with specificity for *S. diaprepesi* named DIAPR1A/DIAPR1B and DIAPR2A/DIAPR2B amplified products of 830 bp and 682 bp, respectively. Sequences obtained from PCR products submitted to sequencing and BLAST showed 100% similarity to *S. diaprepesi* (accession number GU173996.1). The drawn primers did not amplify PCR products from *S. feltiae* and *S. glaseri* populations phylogenetically close to *S. diaprepesi*.

Pathogenicity of S. diaprepesi to G. platensis

Steinernema diaprepesi infected the *G. platensis* pupae, reproduced, and then killed the pupae eight days after inoculation. The three IJ concentrations caused mortality of this host with a high emergence of nematodes ($p < 0.05$) that quickly disintegrated the host pupae. The mortality of this host at the highest concentrations (1.000 and 5.000) of IJs was 100%, whereas no pupae of *G. platensis* died in the control without emergence of nematodes (Table 2).

5. Discussion

The identification of the nematode from molecular analyses with 100% similarity to *S. diaprepesi* is accurate, reliable and safe due to the presence of band and confirmation of the species by Sanger sequencing. This nematode has been characterized in Argentina (Caccia et al., 2017) and Mexico (Molina-Ochoa et al., 2009), and *Heterorhabditis amazonensis* (Andaló, 2006), *Metarhabditis rainai* (Carta & Osbrink 2005), *Oscheius tipulae* (Lam; Webster, 1971) and *Steinernema rarum* (De Doucet, 1986) have been characterized in Brazil (Brida et al., 2017) using this technique. This nematode was first identified in *Diaprepes abbreviatus* Linnaeus (Coleoptera: Curculionidae) larvae in Florida, USA (Nguyen & Duncan, 2002) and, in South America, only in Venezuela and Argentina (Caccia et al., 2017). *Steinernema diaprepesi* is highly virulent for lepidopteran larvae such as those of *Galleria mellonella* L. (Lepidoptera: Pyralidae) and *Spodoptera frugiperda* JE Smith (Lepidoptera: Noctuidae), due to its association with the symbiotic bacterium *Xenorhabdus doucetiae* Tailliez, Pagès, Ginibra & Boemare, 2006 (Caccia et al., 2017; Del Valle et al., 2014). Detection of *S. diaprepesi* and its virulence contributes to integrated pest management programs in Brazil.

The *S. diaprepesi* identification by primers shows their specificity to those nematodes because they do not amplify the DNA of *S. feltiae* and *S. glaseri*, species similar to the first (Nguyen et al., 2002). Specific primers are reliable for identifying species which are morphologically similar according to nucleotide variations, with high sensitivity and velocity (Kaur et al., 2016). This makes it possible to obtain accurate, reliable and useful results for processing a large number of specimens in a single assay and for identifying the desired species at a reduced cost (Kaur et al., 2016).

The death of *G. platensis* pupae eight days after inoculation with the three IJ concentrations confirms their susceptibility to *S. diaprepesi* and the potential of this nematode for managing this pest, as reported for *G. mellonella* and *S. frugiperda* (Del Valle et al., 2014). The efficacy of entomopathogenic nematodes in biological control depends on their ability to locate the host and their virulence (Gaugler, 1987; Shapiro-Ilan et al., 2002). The *G. platensis* mortality rate is related to the association of *S. diaprepesi* with the symbiotic bacterium *X. doucetiae* (Caccia et al., 2017), which when released into host hemocoel, promoted rapid degradation of the host pupae. The high number of juveniles multiplying in the corpse of the host also accelerates degradation due to competition for food and space, depleting the corpse nutrients, which can reduce the number of nematode generations and consequently the number of emerged IJs (Voss et al., 2009). The *S. diaprepesi* pathogenicity can increase the permanence of this nematode in areas infested with *G. platensis* (Shapiro-Ilan et al., 2006), resulting in higher mortality of this pest with a smaller number of applications of this natural enemy.

6. Conclusion

The collection of *S. diaprepesi* in an area with *G. platensis* and the high virulence of this nematode indicate its potential for the integrated management of this pest. This is important because native EPN species exclude risks associated with the introduction of exotic species.

Data accessibility

The data from the survey is freely available on GenBank at <https://www.ncbi.nlm.nih.gov/nucleotide/MT121972> (Damascena et al., 2020).

Authors' Contributions

A.P.D., V.R.C., M.F.R., A.B.H., C.F.W. and S.R.S.W. performed experiments, analyzed the data and designed experiments, A.P.D., B.M.C.C and J.C.Z. wrote and edited the manuscript. All authors read and approved the final manuscript.

Competing Interests

We have no competing interests.

Funding

We thank to the Brazilian institutions “Conselho Nacional de Desenvolvimento Científico e Tecnológico (CNPq)”, “Coordenação de Aperfeiçoamento de Pessoal de Nível Superior- Brasil (CAPES)”, “Fundação de Amparo à Pesquisa do Estado de Minas Gerais (FAPEMIG)”, “Programa Cooperativo sobre Proteção Florestal/PROTEF do Instituto de Pesquisas e Estudos Florestais/IPEF” and “KOPPERT” for financial support.

Acknowledgments

We thank to David Michael Miller, a professional editor and proofreader and native English speaking, has reviewed and edited this article for structure, grammar, punctuation, spelling, word choice, and readability.

References

- Abd-Elsalam KA. 2003 Bioinformatic tools and guideline for PCR primer design. *African Journal of Biotechnology*, **2**, 91-95. (doi: 10.5897/AJB2003.000-1019)
- Al-Banna L, Ploeg AT, Williamson VM, Kaloshian I. 2004 Discrimination of six *Pratylenchus* species using PCR and species-specific primers. *J. Nematol.* **36**, 142–146.
- Batalla-Carrera L, Morton A, Garcia-Del-Pino F. 2016 Virulence of entomopathogenic nematodes and their symbiotic bacteria against the hazelnut weevil *Curculio nucum*. *J. Appl. Entomol.* **140**, 115–123. (doi: 10.1111/jen.12265)
- Caccia M, Dueñas JR, Del Valle E, Doucet ME, Lax P. 2017 Morphological and molecular characterisation of an isolate of *Steinernema diaprepesi* Nguyen & Duncan, 2002 (Rhabditida: Steinernematidae) from Argentina and identification of its bacterial symbiont. *Syst. Parasitol.* **94**, 111-122. (doi: 10.1007/s11230-016-9683-3)
- Damascena AP, Carvalho VR, Ribeiro MF, Horta AB, Castro BMC, Wilcken CF, Zanuncio JC, Wilcken SS. 2020 *Steinernema diaprepesi* isolate 28S rRNA large subunit ribosomal RNA gene, partial sequence. GenBank: MT121972.1 (<https://www.ncbi.nlm.nih.gov/nucleotide/MT121972>)
- De Brida AL, Rosa JMO, De Oliveira CMG, Castro BMC, Serrão JE, Zanuncio JC, Wilcken SRS. 2017 Entomopathogenic nematodes in agricultural areas in Brazil. *Sci Rep-Uk* **7**, 1-7. (doi: 10.1038/srep45254)
- De Carvalho VR, Wilcken SRS, Wilcken CF, Castro BMC, Soares MA, Zanuncio JC. 2018 Technical and economic efficiency of methods for extracting genomic DNA from *Meloidogyne javanica*. *J Microbiol Meth* **157**, 108-112. (doi: 10.1016/j.mimet.2018.12.022)
- Del Valle EE, Balbi EI, Lax P, Dueñas JR, Doucet ME. 2014 Ecological aspects of an isolate of *Steinernema diaprepesi* (Rhabditida: Steinernematidae) from Argentina. *Biocontrol Sci Techn* **24**, 690–704. (doi: 10.1080/09583157.2014.890171)
- Ferreira DF. 2015 **Sisvar**. Versão 5.6. Lavras: UFLA/DEX, Disponible in: <<http://www.dex.ufla.br/~danielff/programas/sisvar.html>>.
- Freeland JR. 2016 The importance of molecular markers and primer design when characterizing biodiversity from environmental DNA. *Genome* **60**, 358-374. (doi: 10.1139/gen-2016-0100)
- Gaugler R. 1987 Entomogenous nematodes and their prospects for genetic improvement. In: Maramorosch K, editor. *Biotechnology in invertebrate pathology and cell culture*. San Diego: Academic Press; pp. 457–484.
- Gumovsky A, De Little D, Rothmann S, Jaques L, Mayorga SE. 2015 Re-description and first host and biology records of *Entedon magnificus* (Girault & Dodd) (Hymenoptera, Eulophidae), a natural enemy of *Gonipterus* weevils (Coleoptera, Curculionidae), a pest of *Eucalyptus* trees. *Zootaxa* **3957**, 577-584. (doi: 10.11646/zootaxa.3957.5.6.)
- Jegger M, Bragard C, Caffier D, Candresse T, Chatzivassiliou E, Dehnen-Schmutz K, Navajas Navarro M. 2018 Pest categorisation of the *Gonipterus scutellatus* species complex. *EFSA Journal* **16**, 1-34. (doi: 10.2903/j.efsa.2018.5107)
- Kaur S, Kang SS, Dhillon NK, Sharma A. 2016 Detection and characterization of *Meloidogyne* species associated with pepper in Indian Punjab. *Nematropica* **46**, 209-220.
- Mapondera TS, Burgess T, Matsuki M, Oberprieler RG. 2012 Identification and molecular phylogenetics of the cryptic species of the *Gonipterus scutellatus* complex (Coleoptera: Curculionidae: Gonipterini). *Aust. J. Entomol.* **51**, 175-188. (doi: 10.1111/j.1440-6055.2011.00853.x)
- Mráček Z, Nguyen KB, Tailler P, Boamare N, Chen S. 2006 *Steinernema sichuanense* n. sp. (Rhabditida, Steinernematidae) a new species of entomopathogenic nematode from the province of Sichuan, east Tibetan Mts., China. *J. Invertebr. Pathol.* **93**, 157-169. (doi: 10.1016/j.jip.2006.06.007)
- Molina-Ochoa J, Nguyen KB, González-Ramires M, Quintana-Moreno MG, Lezama-Gutiérrez R, Foster EF. 2009 *Steinernema diaprepesi* (Nematoda: Steinernematidae): its occurrence in Western Mexico and susceptibility of engorged cattle ticks *Boophilus microplus* (Acari: Ixodidae). *Fla. Entomol.* **92**, 660–663. (doi: 10.1653 / 024.092.0423)
- Nascimento LI, Soliman EP, Zauza EAV, Stape JL, Wilcken CF. 2017 First global record of *Podisus nigrispinus* (Hemiptera: Pentatomidae) as predator of *Gonipterus platensis* (Coleoptera: Curculionidae) larvae and adults. *Fla Entomol.* **100**, 675-677. (doi: 10.1653 / 024.100.0331)

- 1 Nguyen KB, Duncan LW. 2002 *Steinernema diaprepesi* n. sp. (Rhabditida: Steinernematidae), a parasite of the citrus root
weevil *Diaprepes abbreviatus* (L) (Coleoptera: Curculionidae). *J. Nematol.* **34**, 159.
- 2 Nguyen KB, Maruniak J, Adams BJ. 2001 Diagnostic and phylogenetic utility of the rDNA internal transcribed spacer
3 sequences of *Steinernema*. *J. Nematol.* **33**, 73–82.
- 4 Poinar GO. Biology and taxonomy of Steinernematidae and Heterorhabditidae. In: Gaugler R, Kaya HK. 1990
5 Entomopathogenic nematodes in biological control. Boca Raton, FL: CRC Press, 23–62.
- 6 Reis AR, Ferreira L, Tomé M, Araujo C, Branco M. 2012 Efficiency of biological control of *Gonipterus platensis* (Coleoptera:
7 Curculionidae) by *Anaphes nitens* (Hymenoptera: Mymaridae) in cold areas of the Iberian Peninsula: Implications
8 for defoliation and wood production in *Eucalyptus globulus*. *Forest Ecol. Manag.* **270**, 216–222. (doi:
9 10.1016/j.foreco.2012.01.038)
- 10 Shapiro-Ilan DI, Stuart RJ, McCoy CW. 2006. A comparison of entomopathogenic nematode longevity in soil under
11 laboratory conditions. *J. Nematol.* **38**, 119.
- 12 Shapiro-Ilan DI, Gouge DH, Koppenhöfer AM. 2002 Factors affecting commercial success: Case studies in cotton, turf, and
13 citrus. In: Gaugler R, editor. Entomopathogenic nematology. New York: CABI; 333–356.
- 14 Souza NM, Junqueira RL, Wilcken CF, Soliman EP, Camargo MB, Nিকেle MA, Barbosa LR. 2016 Ressurgência de uma
15 antiga ameaça: Gorgulho-do-eucalipto *Gonipterus platensis* (Coleoptera: Curculionidae). *Circular Técnica* 209.
16 Piracicaba: Instituto de Pesquisas e Estudos Florestais, 20 p.
- 17 Tarasco E, Clausi M, Rappazzo G, Panzavolta T, Curto G, Sorino R, Vinciguerra MT. 2015 Biodiversity of
18 entomopathogenic nematodes in Italy. *J. Helminthol.* **89**, 359–366. (doi: 10.1017 / S0022149X14000194)
- 19 Tumialis D, Pezowicz E, Skrzecz I, Mazurkiewicz A, Maszewska J, Pietraszczyk JJ, Kucharska K. 2016 Occurrence of
20 entomopathogenic nematodes in Polish soils. *Cienc. Rural* **46**, 1126–1129. (doi: 10.1590/0103-8478cr20151542)
- 21 Valente C, Gonçalves CI, Monteiro F, Gaspar J, Silva M, Sottomayor M, Branco M. 2018 Economic outcome of classical
22 biological Control: A case study on the eucalyptus Snout Beetle, *Gonipterus platensis*, and the parasitoid *Anaphes*
23 *nitens*. *Ecol. Econ.* **149**, 40–47. (doi: 10.1016/j.ecolecon.2018.03.001)
- 24 Valente C, Gonçalves CI, Reis A, Branco M. 2017 Pre-selection and biological potential of the egg parasitoid *Anaphes*
25 *inexpectatus* for the control of the Eucalyptus snout beetle, *Gonipterus platensis*. *J. Pest Sci.* **90**, 911–923.
- 26 Voss M, Andaló V, Negrisoni Júnior AS, Barbosa-Negrisoni CR. 2009 Manual de técnicas laboratoriais para obtenção,
27 manutenção e caracterização de nematoides entomopatogênicos. Embrapa Trigo- Documentos (INFOTECA-E).
- 28 White GF. 1927 A method for obtaining infective nematode larvae from cultures. *Science* **66**, 302–303.
- 29 Williams BD, Schrank B, Huynh C, Shownkeen R, Waterston RH. 1992 A genetic mapping system in *Caenorhabditis elegans*
30 based on polymorphic sequence-tagged sites. *Genetics* **131**, 609–624.
- 31
32
33
34
35
36
37
38
39
40
41
42
43
44
45
46
47
48
49
50
51
52
53
54
55
56
57
58
59
60

Tables

Table 1. Primers used to amplify the genomic DNA of entomopathogenic nematodes obtained from soil in an area with eucalyptus plantation

Primers	Direction	Sequence
D2A ¹	Forward	CAAGTACCGTGAGGGAAAGTTG
D3B ¹	Reverse	TCGGAAGGAACCAG CTACTA
KN58 ²	Forward	GTATGTTTGGTTGAAGGTC
KNRV ²	Reverse	CACGCTCATACAACTGCTC
DIAPR1A	Forward	CGTAGGTGAACCTGCGGAAG
DIAPR1B	Reverse	G TTCAGCGGGTAGTCTTGCT
DIAPR2A	Forward	ACTGCTTCTCTGAGCGCTTT
DIAPR2B	Reverse	CCTCCATTAGCCCATCGCAT

¹ Al-Banna et al., 2004; ² Nguyen et al., 2001

Table 2. Mortality of *Gonipterus platensis* (Coleoptera: Curculionidae) and number of individuals (mean ± standard error and range) of *Steinernema diaprepesi* (Rhabditida: Steinernematidae) emerged per pupa of this insect eight days after inoculation with different concentrations of infective juveniles (IJs)

Treatments	Mortality (%)	Emergence
Control	0 ± 0 a	0 ± 0 a
500 IJs	80 ± 13.3b	33735 ± 7913b
1000 IJs	100 ± 0.0b	35625 ± 6556b
5000 IJs	100 ± 0.0b	32920 ± 6523b
CV (%)	31.51	51.72

Means followed by the same letter, per column, did not differ at 5% probability level by the Tukey test (p<0.05).

Table captions

Table 1. Primers used to amplify the genomic DNA of entomopathogenic nematodes obtained from soil in an area with eucalyptus plantation

Table 2. Mortality of *Gonipterus platensis* (Coleoptera: Curculionidae) and number of individuals (mean ± standard error and range) of *Steinernema diaprepesi* (Rhabditida: Steinernematidae) emerged per pupa of this insect eight days after inoculation with different concentrations of infective juveniles (IJs)

Appendix B**ROYAL SOCIETY
OPEN SCIENCE****Steinernema diaprepesi (Rhabditida: Steinernematidae)
parasitizing Gonipterus platensis (Coleoptera:
Curculionidae)**

Journal:	Royal Society Open Science
Manuscript ID	RSOS-200282
Article Type:	Research
Date Submitted by the Author:	06-Mar-2020
Complete List of Authors:	Damascena, Aixelhe; Universidade Estadual Paulista, Departamento de Proteção Vegetal Carvalho, Vanessa; Universidade Estadual Paulista, Departamento de Proteção Vegetal Ribeiro, Murilo; Universidade Estadual Paulista, Departamento de Proteção Vegetal Horta, André; Universidade Estadual Paulista, Departamento de Proteção Vegetal Castro, Bárbara; Universidade Federal de Viçosa, Wilcken, Carlos; Universidade Estadual Paulista, Departamento de Proteção Vegetal Zanuncio, José; Universidade Federal de Viçosa, Departamento de Biologia Animal; Wilcken, Sílvia; Universidade Estadual Paulista, Departamento de Proteção Vegetal
Subject:	ecology < BIOLOGY
Keywords:	Biological control, eucalyptus weevil, entomopathogenic nematodes, pathogenicity
Subject Category:	Organismal and Evolutionary Biology

Author-supplied statements

Relevant information will appear here if provided.

Ethics

Does your article include research that required ethical approval or permits?:

This article does not present research with ethical considerations

Statement (if applicable):

CUST_IF_YES_ETHICS :No data available.

Data

It is a condition of publication that data, code and materials supporting your paper are made publicly available. Does your paper present new data?:

Yes

Statement (if applicable):

The data from the survey is freely available on GenBank at <https://www.ncbi.nlm.nih.gov/nuccore/MT121972> (Damascena et al., 2020)

Conflict of interest

I/We declare we have no competing interests

Statement (if applicable):

CUST_STATE_CONFLICT :No data available.

Authors' contributions

This paper has multiple authors and our individual contributions were as below

Statement (if applicable):

A.P.D., V.R.C., M.F.R., A.B.H., C.F.W. and S.R.S.W. performed experiments, analyzed the data and designed experiments, A.P.D., B.M.C.C and J.C.Z. wrote and edited the manuscript. All authors read and approved the final manuscript.

***Steinernema diaprepesi* (Rhabditida: Steinernematidae) parasitizing *Gonipterus platensis* (Coleoptera: Curculionidae)**

Alixelhe Pacheco Damascena¹; Vanessa Rafaela de Carvalho¹, Murilo Fonseca Ribeiro¹, André Ballerini Horta¹, Bárbara Monteiro de Castro e Castro², Carlos Frederico Wilcken¹, José Cola Zanuncio², Silvia Renata Siciliano Wilcken¹

¹ Faculdade de Ciências Agronômicas, Departamento de Proteção Vegetal, Universidade Estadual Paulista (UNESP), 18610-034, Botucatu, São Paulo, Brasil.

² Departamento de Entomologia/BIOAGRO, Universidade Federal de Viçosa, 36570-900, Viçosa, Minas Gerais, Brasil.

Keywords: Biological control, eucalyptus weevil, entomopathogenic nematodes, pathogenicity.

1. Summary

Entomopathogenic nematodes (EPNs) can control pests due to the mutualistic association with bacteria. The objective of this study was to identify an EPN isolate collected in eucalyptus cultivation and to determine its pathogenicity with regard to *Gonipterus platensis*. Four steel-mesh traps with two seventh-instar *Galleria mellonella* larvae were buried 5 cm deep in the soil in a commercial *Eucalyptus* plantation. After seven days, the traps were packed in plastic bags and transported to laboratory to isolate the EPNs using White traps. The obtained nematodes were multiplied in *G. mellonella* larvae and identified by sequencing their D2/D3 expansion of the 28S rDNA region by PCR and specific primers for ITS regions. *Steinernema diaprepesi* was identified and inoculated into *G. platensis* pupae at doses of 500, 1,000 and 5,000 infective juveniles (IJs) to determine its pathogenicity to this pest. At eight days after inoculation, the mortality rate of the *G. platensis* pupae was 80% with the lowest concentration and 100% with the others. The emergence of nematodes and the rapid degradation of *G. platensis* pupae were observed in those inoculated with IJs. The pathogenicity to the *G. platensis* pupae indicates potential for using this nematode in the integrated management of this insect.

2. Introduction

Gonipterus platensis Marelli (Coleoptera: Curculionidae), the eucalyptus weevil, native to Tasmania, is one of the major *Eucalyptus* pests. This beetle was reported in New Zealand in 1890 and causes crop damage in South America, southwestern North America (California), the Iberian Peninsula, the Canary Islands and Hawaii (Mapondera et al., 2012). The attack severity and high dispersal capacity justify efforts to manage this pest (Jegger et al., 2018). Adults and, mainly, *G. platensis* larvae which feed exclusively on young eucalyptus leaves can damage this plant. Severe attacks and successive defoliation can cause shoot death, reduced growth, bifurcation and warping of the plant stem (Souza et al., 2016).

Biological control with the *Anaphes nitens* Girault (Hymenoptera: Mymaridae) egg parasitoid, native to Australia, is the main management strategy for the eucalyptus weevil (Jegger et al., 2018). The control efficiency of this pest is low in many regions, due to factors such as altitude and parasitoid-host

*Author for correspondence (barbaramcastro@hotmail.com).

†Present address: Departamento de Entomologia/BIOAGRO, Universidade Federal de Viçosa, Viçosa, Minas Gerais, 36570-900, Brasil.

incompatibility (Mapondera et al., 2012; Reis et al., 2012). However, losses caused by the eucalyptus weevil are two to three times lower with biological control than with other management strategies, such as replacement of genetic material and use of insecticides (Valente et al., 2018).

The identification of natural enemies acting together with *A. nitens*, can increase the control efficiency of *Gonipterus* spp. In general, this search has been focused on natural enemies of the egg, larva and adult stages of this pest, present in the upper third of the trees (Gumovsky et al., 2015; Nascimento et al., 2017; Valente et al., 2017), while control of the pupae of this pest, which occurs in the soil, has been deficient.

Entomopathogenic nematodes (EPNs) (Steinernematidae and Heterorhabditidae) are effective in pest management due to their association with bacteria of the genus *Xenorhabdus* (Thomas, Poinar) and *Photorhabdus* (Boemare; Louis, Kuhl) (Batalla-Carrera et al., 2016.). Infective juveniles (IJs) of these nematodes release the bacteria into the hemocele, killing the host by septicemia in 24 to 48 hours, and making the environment favorable for its development and reproduction (Poinar, 1990).

Population surveys of entomopathogenic nematodes (EPNs), that inhabit the soil of different environments (Tarasco et al., 2015; Tumialis et al., 2016), are important for identifying their species and efficacy in biological control programs (De Brida et al., 2017).

The objective of this study was to identify EPN species in eucalyptus cultivation and to evaluate their pathogenicity for *G. platensis*.

3. Materials and Methods

Nematode Survey

Four steel-mesh traps with two seventh instars of *Galleria mellonella* L. (Lepidoptera: Pyralidae) larvae each were buried in the soil at a depth of 5 cm in a commercial *Eucalyptus grandis* × *Eucalyptus urophylla* plantation where *G. platensis* larvae and adults were present. After seven days, the traps were removed from the soil, packed in plastic bags and transported to the FCA/UNESP Nematology Laboratory for isolation and identification of entomopathogenic nematodes.

Galleria mellonella larvae were removed from the traps, washed in sodium hypochlorite solution (1%) and transferred to White traps (White, 1927) in an incubator (B.O.D.) at 25 °C for 21 days. Infective juveniles (IJs) obtained from these traps were inoculated in *G. mellonella* larvae to identify the nematode.

Multiplication of the Isolate

Infectious juveniles (IJs) were multiplied in *G. mellonella* larvae. Five insect larvae were placed per Petri dish (9 cm in diameter × 1.5 cm in height), which were coated with filter paper moistened with a nematode suspension at a concentration of 500 IU/cm². Dead larvae were transferred to White traps (White, 1927) and stored in an incubator (B.O.D.) at 25 °C for 21 days. The IJs obtained from these hosts were collected in water film (1 cm deep) in Erlemeyers kept in an incubation chamber (B.O.D.) at a temperature of 18 °C and 70% RH.

Molecular Identification

The genomic DNA of 50 IJs, isolated from the samples, was extracted in 50 µl of 0.85% NaCl with the modified WormLysis Buffer (WLB) extraction method (Carvalho et al., 2018; Williams et al., 1992). DNA samples from EPNs were plated with KCl (50 mM), Tris (10 mM pH 8.2), MgCl₂ (2.5 mM), Tween 20 (0.45%) and proteinase K (20 mg/ml) without gelatin (0.05%). These samples were left at -70 °C for 15 minutes, incubated for one hour at 60 °C, then 15 minutes at 95 °C and stored at -20 °C.

The extracted DNA was amplified with the Polymerase Chain Reaction (PCR) technique with the universal primers D2A and D3B for the expansion of 28S rDNA sequence (Al-Banna et al., 2004), internal to the ITS, KN58 and KNRV (Nguyen et al., 2001) and the specific drawn (Table 1). The reactions were performed in an INFINIGEN Thermal Cycler (model TC-96CG) in 0.5 ml tubes with 12.5 µl of Polymerase Mix Master Red (Neobio), 7.5 µl of NucleaseFreeWater (Promega), 1 µl of each primer (10 mM) and 3 µl of genomic DNA per sample, totaling 25 µl of solution per tube and reaction. The cycles for the universal primers D2A and D3B were preceded by initial denaturation at 94 °C for seven minutes, followed by 35 cycles of denaturation at 94 °C for one minute, annealing at 55 °C for one minute, extension at 72 °C for one minute and final extension of 72 °C for ten minutes (Mracek et al., 2006). The primers KN58 and KNRV were amplified by PCR according to the established cycling (Nguyen et al., 2001), and those drawn were preceded by initial denaturation at 94 °C for seven minutes, followed by 35 denaturation cycles at 94 °C for one minute,

annealing at 58 °C for one minute, extension at 72 °C for one minute and final extension of 72 °C for ten minutes. Negative controls with 3µl of water were added in the assays to check for possible PCR reagent contaminations. PCR amplification products were visualized by 1% agarose gel electrophoresis with 100 bp marker (Norgen) and UV Light Transilluminator (Major Science).

The PCR product was purified according to recommendations for the Celcco PCR Purification Kit (Qiagen, Cat # 14400) and sequenced in a Sanger Automated DNA Sequencer (Model: ABI 3500- Applied Biosystems) at the Institute of Biotechnology (IBTEC-UNESP). The species was identified by the sequences obtained, which were aligned and compared in the BLAST with the data deposited in GenBank (<http://www.ncbi.nlm.nih.gov>). Specific primers, based on the genomic sequence obtained, were designed after sequencing and tested in populations of *Steinernema feltiae* (Filipjev) and *Steinernema glaseri* (Steiner), phylogenetically close to *S. diaprepesi* (Nguyen & Duncan, 2002), to prove their specificity. These primers were annealed at 50-60 °C with 40% to 60% guanine and cytosine without formation of secondary structures, region 18 to 22 nucleotides and PCR products between 500 and 1200 bp (Freeland, 2016, Abd-Elsalam et al., 2003).

Obtaining *Gonipterus platensis*

Gonipterus platensis pupae were obtained from the Laboratory of Biological Control of Forest Pests (LCBPF) of the Faculdade de Agronomia of the Universidade Estadual Paulista (FCA/UNESP). This insect was created in an air-conditioned room at 25 ± 1 °C, relative humidity of 50 ± 10% and photoperiod 12: 12h (L: D). Adults of these insects were kept in wooden cages (40 x 45 x 80 cm) covered with *voil* and fed with *Eucalyptus urophylla* leaves, where they oviposited. The *G. platensis* larvae were fed with tender *Eucalyptus urophylla* leaves in branch bundles.

Pathogenicity of the Entomopathogenic Nematode to *G. platensis*

Gonipterus platensis pre-pupae were individualized in 50 ml plastic pots with 32 g of autoclaved and sieved sand and 3 ml of distilled water, mixed with the aid of a sterile rod. After 15 days, 2 ml of the suspension with 500, 1000 or 5000 IU of 48-hour-old EPNs were added to the substrate. Pupae in the control received 2 mL of distilled water.

The plastic pots with *G. platensis* pupae were kept in an incubator (B.O.D.) at 25 ± 2 °C and 70 ± 10% RH. Mortality of these pupae was evaluated eight days after inoculation of the nematode. Dead pupae were individualized in White traps (White, 1927) for the emergence of the nematodes.

The experiment was conducted according to a completely randomized design, with four treatments and 10 replications, with one with *G. platensis* pre-pupae. The data were analyzed using the statistical software SISVAR 5.6 and the means compared by the Tukey test at 5% significance (Ferreira, 2015).

4. Results

Molecular Identification of the Nematoid

The nucleotide sequences obtained from the Sanger sequencing showed 100% similarity to those of *Steinernema diaprepesi* (Rhabditida: Steinernematidae) (accession number GU173994.1).

Steinernema diaprepesi was identified with primers KN58 and KNRV of the ITS region, which did not amplify DNA fragments of this nematode. Pairs of specific primers designed with specificity for *S. diaprepesi* named DIAPR1A/DIAPR1B and DIAPR2A/DIAPR2B amplified products of 830 bp and 682 bp, respectively. Sequences obtained from PCR products submitted to sequencing and BLAST showed 100% similarity to *S. diaprepesi* (accession number GU173996.1). The drawn primers did not amplify PCR products from *S. feltiae* and *S. glaseri* populations phylogenetically close to *S. diaprepesi*.

Pathogenicity of S. diaprepesi to G. platensis

Steinernema diaprepesi infected the *G. platensis* pupae, reproduced, and then killed the pupae eight days after inoculation. The three IJ concentrations caused mortality of this host with a high emergence of nematodes ($p < 0.05$) that quickly disintegrated the host pupae. The mortality of this host at the highest concentrations (1.000 and 5.000) of IJs was 100%, whereas no pupae of *G. platensis* died in the control without emergence of nematodes (Table 2).

5. Discussion

The identification of the nematode from molecular analyses with 100% similarity to *S. diaprepesi* is accurate, reliable and safe due to the presence of band and confirmation of the species by Sanger sequencing. This nematode has been characterized in Argentina (Caccia et al., 2017) and Mexico (Molina-Ochoa et al., 2009), and *Heterorhabditis amazonensis* (Andaló, 2006), *Metarhabditis rainai* (Carta & Osbrink 2005), *Oscheius tipulae* (Lam; Webster, 1971) and *Steinernema rarum* (De Doucet, 1986) have been characterized in Brazil (Brida et al., 2017) using this technique. This nematode was first identified in *Diaprepes abbreviatus* Linnaeus (Coleoptera: Curculionidae) larvae in Florida, USA (Nguyen & Duncan, 2002) and, in South America, only in Venezuela and Argentina (Caccia et al., 2017). *Steinernema diaprepesi* is highly virulent for lepidopteran larvae such as those of *Galleria mellonella* L. (Lepidoptera: Pyralidae) and *Spodoptera frugiperda* JE Smith (Lepidoptera: Noctuidae), due to its association with the symbiotic bacterium *Xenorhabdus doucetiae* Tailliez, Pagès, Ginibra & Boemare, 2006 (Caccia et al., 2017; Del Valle et al., 2014). Detection of *S. diaprepesi* and its virulence contributes to integrated pest management programs in Brazil.

The *S. diaprepesi* identification by primers shows their specificity to those nematodes because they do not amplify the DNA of *S. feltiae* and *S. glaseri*, species similar to the first (Nguyen et al., 2002). Specific primers are reliable for identifying species which are morphologically similar according to nucleotide variations, with high sensitivity and velocity (Kaur et al., 2016). This makes it possible to obtain accurate, reliable and useful results for processing a large number of specimens in a single assay and for identifying the desired species at a reduced cost (Kaur et al., 2016).

The death of *G. platensis* pupae eight days after inoculation with the three IJ concentrations confirms their susceptibility to *S. diaprepesi* and the potential of this nematode for managing this pest, as reported for *G. mellonella* and *S. frugiperda* (Del Valle et al., 2014). The efficacy of entomopathogenic nematodes in biological control depends on their ability to locate the host and their virulence (Gaugler, 1987; Shapiro-Ilan et al., 2002). The *G. platensis* mortality rate is related to the association of *S. diaprepesi* with the symbiotic bacterium *X. doucetiae* (Caccia et al., 2017), which when released into host hemocoel, promoted rapid degradation of the host pupae. The high number of juveniles multiplying in the corpse of the host also accelerates degradation due to competition for food and space, depleting the corpse nutrients, which can reduce the number of nematode generations and consequently the number of emerged IJs (Voss et al., 2009). The *S. diaprepesi* pathogenicity can increase the permanence of this nematode in areas infested with *G. platensis* (Shapiro-Ilan et al., 2006), resulting in higher mortality of this pest with a smaller number of applications of this natural enemy.

6. Conclusion

The collection of *S. diaprepesi* in an area with *G. platensis* and the high virulence of this nematode indicate its potential for the integrated management of this pest. This is important because native EPN species exclude risks associated with the introduction of exotic species.

Data accessibility

The data from the survey is freely available on GenBank at <https://www.ncbi.nlm.nih.gov/nucleotide/MT121972> (Damascena et al., 2020).

Authors' Contributions

A.P.D., V.R.C., M.F.R., A.B.H., C.F.W. and S.R.S.W. performed experiments, analyzed the data and designed experiments, A.P.D., B.M.C.C and J.C.Z. wrote and edited the manuscript. All authors read and approved the final manuscript.

Competing Interests

We have no competing interests.

Funding

We thank to the Brazilian institutions “Conselho Nacional de Desenvolvimento Científico e Tecnológico (CNPq)”, “Coordenação de Aperfeiçoamento de Pessoal de Nível Superior- Brasil (CAPES)”, “Fundação de Amparo à Pesquisa do Estado de Minas Gerais (FAPEMIG)”, “Programa Cooperativo sobre Proteção Florestal/PROTEF do Instituto de Pesquisas e Estudos Florestais/IPEF” and “KOPPERT” for financial support.

Acknowledgments

We thank to David Michael Miller, a professional editor and proofreader and native English speaking, has reviewed and edited this article for structure, grammar, punctuation, spelling, word choice, and readability.

References

- Abd-Elsalam KA. 2003 Bioinformatic tools and guideline for PCR primer design. *African Journal of Biotechnology*, **2**, 91-95. (doi: 10.5897/AJB2003.000-1019)
- Al-Banna L, Ploeg AT, Williamson VM, Kaloshian I. 2004 Discrimination of six *Pratylenchus* species using PCR and species-specific primers. *J. Nematol.* **36**, 142–146.
- Batalla-Carrera L, Morton A, Garcia-Del-Pino F. 2016 Virulence of entomopathogenic nematodes and their symbiotic bacteria against the hazelnut weevil *Curculio nucum*. *J. Appl. Entomol.* **140**, 115–123. (doi: 10.1111/jen.12265)
- Caccia M, Dueñas JR, Del Valle E, Doucet ME, Lax P. 2017 Morphological and molecular characterisation of an isolate of *Steinernema diaprepesi* Nguyen & Duncan, 2002 (Rhabditida: Steinernematidae) from Argentina and identification of its bacterial symbiont. *Syst. Parasitol.* **94**, 111-122. (doi: 10.1007/s11230-016-9683-3)
- Damascena AP, Carvalho VR, Ribeiro MF, Horta AB, Castro BMC, Wilcken CF, Zanuncio JC, Wilcken SS. 2020 *Steinernema diaprepesi* isolate 28S rRNA large subunit ribosomal RNA gene, partial sequence. GenBank: MT121972.1 (<https://www.ncbi.nlm.nih.gov/nucleotide/MT121972>)
- De Brida AL, Rosa JMO, De Oliveira CMG, Castro BMC, Serrão JE, Zanuncio JC, Wilcken SRS. 2017 Entomopathogenic nematodes in agricultural areas in Brazil. *Sci Rep-Uk* **7**, 1-7. (doi: 10.1038/srep45254)
- De Carvalho VR, Wilcken SRS, Wilcken CF, Castro BMC, Soares MA, Zanuncio JC. 2018 Technical and economic efficiency of methods for extracting genomic DNA from *Meloidogyne javanica*. *J Microbiol Meth* **157**, 108-112. (doi: 10.1016/j.mimet.2018.12.022)
- Del Valle EE, Balbi EI, Lax P, Dueñas JR, Doucet ME. 2014 Ecological aspects of an isolate of *Steinernema diaprepesi* (Rhabditida: Steinernematidae) from Argentina. *Biocontrol Sci Techn* **24**, 690–704. (doi: 10.1080/09583157.2014.890171)
- Ferreira DF. 2015 **Sisvar**. Versão 5.6. Lavras: UFLA/DEX, Disponible in: <<http://www.dex.ufla.br/~danielff/programas/sisvar.html>>.
- Freeland JR. 2016 The importance of molecular markers and primer design when characterizing biodiversity from environmental DNA. *Genome* **60**, 358-374. (doi: 10.1139/gen-2016-0100)
- Gaugler R. 1987 Entomogenous nematodes and their prospects for genetic improvement. In: Maramorosch K, editor. *Biotechnology in invertebrate pathology and cell culture*. San Diego: Academic Press; pp. 457–484.
- Gumovsky A, De Little D, Rothmann S, Jaques L, Mayorga SE. 2015 Re-description and first host and biology records of *Entedon magnificus* (Girault & Dodd) (Hymenoptera, Eulophidae), a natural enemy of *Gonipterus* weevils (Coleoptera, Curculionidae), a pest of *Eucalyptus* trees. *Zootaxa* **3957**, 577-584. (doi: 10.11646/zootaxa.3957.5.6.)
- Jegger M, Bragard C, Caffier D, Candresse T, Chatzivassiliou E, Dehnen-Schmutz K, Navajas Navarro M. 2018 Pest categorisation of the *Gonipterus scutellatus* species complex. *EFSA Journal* **16**, 1-34. (doi: 10.2903/j.efsa.2018.5107)
- Kaur S, Kang SS, Dhillon NK, Sharma A. 2016 Detection and characterization of *Meloidogyne* species associated with pepper in Indian Punjab. *Nematropica* **46**, 209-220.
- Mapondera TS, Burgess T, Matsuki M, Oberprieler RG. 2012 Identification and molecular phylogenetics of the cryptic species of the *Gonipterus scutellatus* complex (Coleoptera: Curculionidae: Gonipterini). *Aust. J. Entomol.* **51**, 175-188. (doi: 10.1111/j.1440-6055.2011.00853.x)
- Mráček Z, Nguyen KB, Tailler P, Boamare N, Chen S. 2006 *Steinernema sichuanense* n. sp. (Rhabditida, Steinernematidae) a new species of entomopathogenic nematode from the province of Sichuan, east Tibetan Mts., China. *J. Invertebr. Pathol.* **93**, 157-169. (doi: 10.1016/j.jip.2006.06.007)
- Molina-Ochoa J, Nguyen KB, González-Ramires M, Quintana-Moreno MG, Lezama-Gutiérrez R, Foster EF. 2009 *Steinernema diaprepesi* (Nematoda: Steinernematidae): its occurrence in Western Mexico and susceptibility of engorged cattle ticks *Boophilus microplus* (Acari: Ixodidae). *Fla. Entomol.* **92**, 660–663. (doi: 10.1653 / 024.092.0423)
- Nascimento LI, Soliman EP, Zauza EAV, Stape JL, Wilcken CF. 2017 First global record of *Podisus nigrispinus* (Hemiptera: Pentatomidae) as predator of *Gonipterus platensis* (Coleoptera: Curculionidae) larvae and adults. *Fla Entomol.* **100**, 675-677. (doi: 10.1653 / 024.100.0331)

- 1 Nguyen KB, Duncan LW. 2002 *Steinernema diaprepesi* n. sp. (Rhabditida: Steinernematidae), a parasite of the citrus root
weevil *Diaprepes abbreviatus* (L) (Coleoptera: Curculionidae). *J. Nematol.* **34**, 159.
- 2 Nguyen KB, Maruniak J, Adams BJ. 2001 Diagnostic and phylogenetic utility of the rDNA internal transcribed spacer
3 sequences of *Steinernema*. *J. Nematol.* **33**, 73–82.
- 4 Poinar GO. Biology and taxonomy of Steinernematidae and Heterorhabditidae. In: Gaugler R, Kaya HK. 1990
5 Entomopathogenic nematodes in biological control. Boca Raton, FL: CRC Press, 23–62.
- 6 Reis AR, Ferreira L, Tomé M, Araujo C, Branco M. 2012 Efficiency of biological control of *Gonipterus platensis* (Coleoptera:
7 Curculionidae) by *Anaphes nitens* (Hymenoptera: Mymaridae) in cold areas of the Iberian Peninsula: Implications
8 for defoliation and wood production in *Eucalyptus globulus*. *Forest Ecol. Manag.* **270**, 216–222. (doi:
9 10.1016/j.foreco.2012.01.038)
- 10 Shapiro-Ilan DI, Stuart RJ, McCoy CW. 2006. A comparison of entomopathogenic nematode longevity in soil under
11 laboratory conditions. *J. Nematol.* **38**, 119.
- 12 Shapiro-Ilan DI, Gouge DH, Koppenhöfer AM. 2002 Factors affecting commercial success: Case studies in cotton, turf, and
13 citrus. In: Gaugler R, editor. Entomopathogenic nematology. New York: CABI; 333–356.
- 14 Souza NM, Junqueira RL, Wilcken CF, Soliman EP, Camargo MB, Nিকেle MA, Barbosa LR. 2016 Ressurgência de uma
15 antiga ameaça: Gorgulho-do-eucalipto *Gonipterus platensis* (Coleoptera: Curculionidae). *Circular Técnica* 209.
16 Piracicaba: Instituto de Pesquisas e Estudos Florestais, 20 p.
- 17 Tarasco E, Clausi M, Rappazzo G, Panzavolta T, Curto G, Sorino R, Vinciguerra MT. 2015 Biodiversity of
18 entomopathogenic nematodes in Italy. *J. Helminthol.* **89**, 359–366. (doi: 10.1017 / S0022149X14000194)
- 19 Tumialis D, Pezowicz E, Skrzecz I, Mazurkiewicz A, Maszewska J, Pietraszczyk JJ, Kucharska K. 2016 Occurrence of
20 entomopathogenic nematodes in Polish soils. *Cienc. Rural* **46**, 1126–1129. (doi: 10.1590/0103-8478cr20151542)
- 21 Valente C, Gonçalves CI, Monteiro F, Gaspar J, Silva M, Sottomayor M, Branco M. 2018 Economic outcome of classical
22 biological Control: A case study on the eucalyptus Snout Beetle, *Gonipterus platensis*, and the parasitoid *Anaphes*
23 *nitens*. *Ecol. Econ.* **149**, 40–47. (doi: 10.1016/j.ecolecon.2018.03.001)
- 24 Valente C, Gonçalves CI, Reis A, Branco M. 2017 Pre-selection and biological potential of the egg parasitoid *Anaphes*
25 *inexpectatus* for the control of the Eucalyptus snout beetle, *Gonipterus platensis*. *J. Pest Sci.* **90**, 911–923.
- 26 Voss M, Andaló V, Negrisoni Júnior AS, Barbosa-Negrisoni CR. 2009 Manual de técnicas laboratoriais para obtenção,
27 manutenção e caracterização de nematoides entomopatogênicos. Embrapa Trigo- Documentos (INFOTECA-E).
- 28 White GF. 1927 A method for obtaining infective nematode larvae from cultures. *Science* **66**, 302–303.
- 29 Williams BD, Schrank B, Huynh C, Shownkeen R, Waterston RH. 1992 A genetic mapping system in *Caenorhabditis elegans*
30 based on polymorphic sequence-tagged sites. *Genetics* **131**, 609–624.
- 31
32
33
34
35
36
37
38
39
40
41
42
43
44
45
46
47
48
49
50
51
52
53
54
55
56
57
58
59
60

Tables

Table 1. Primers used to amplify the genomic DNA of entomopathogenic nematodes obtained from soil in an area with eucalyptus plantation

Primers	Direction	Sequence
D2A ¹	Forward	CAAGTACCGTGAGGGAAAGTTG
D3B ¹	Reverse	TCGGAAGGAACCAG CTACTA
KN58 ²	Forward	GTATGTTTGGTTGAAGGTC
KNRV ²	Reverse	CACGCTCATACAACTGCTC
DIAPR1A	Forward	CGTAGGTGAACCTGCCGAAG
DIAPR1B	Reverse	G TTCAGCGGGTAGTCTTGCT
DIAPR2A	Forward	ACTGCTTCTCTGAGCGCTTT
DIAPR2B	Reverse	CCTCCATTAGCCCATCGCAT

¹ Al-Banna et al., 2004; ² Nguyen et al., 2001

Table 2. Mortality of *Gonipterus platensis* (Coleoptera: Curculionidae) and number of individuals (mean ± standard error and range) of *Steinernema diaprepesi* (Rhabditida: Steinernematidae) emerged per pupa of this insect eight days after inoculation with different concentrations of infective juveniles (IJs)

Treatments	Mortality (%)	Emergence
Control	0 ± 0 a	0 ± 0 a
500 IJs	80 ± 13.3b	33735 ± 7913b
1000 IJs	100 ± 0.0b	35625 ± 6556b
5000 IJs	100 ± 0.0b	32920 ± 6523b
CV (%)	31.51	51.72

Means followed by the same letter, per column, did not differ at 5% probability level by the Tukey test (p<0.05).

Table captions

Table 1. Primers used to amplify the genomic DNA of entomopathogenic nematodes obtained from soil in an area with eucalyptus plantation

Table 2. Mortality of *Gonipterus platensis* (Coleoptera: Curculionidae) and number of individuals (mean ± standard error and range) of *Steinernema diaprepesi* (Rhabditida: Steinernematidae) emerged per pupa of this insect eight days after inoculation with different concentrations of infective juveniles (IJs)

Appendix C

May, 18th 2019

Dear Editor, Andrew Dunn:

Please find enclosed our revised manuscript entitled “*Steinernema diaprepesi* (Rhabditida: Steinernematidae) parasitizing *Gonipterus platensis* (Coleoptera: Curculionidae)” (RSOS-200282). The main changes are red-marked in the manuscript, and the point-by-point explanations to the comments provided as follows:

COMMENTS FROM EDITORS AND REVIEWERS

Editor comments:

One reviewer is quite favorable and offers useful comments. The other is less so but it seems that these could be relatively easily addressed because they don't question the main science of the paper. Please address all comments and best wishes for revising.

The comments and suggestion provided were greatly appreciated and carefully considered for preparing the current version.

Comments to Author:

Reviewers' Comments to Author:

Reviewer: 1

Comments to the Author(s)

The manuscript is interesting, well written and aims identify an EPN isolate collected in eucalyptus cultivation and to determine its pathogenicity with regard to *Gonipterus platensis*. This insect is an important pest worldwide and the identification of the pathogen may increase the IPM tools for this pest. The information can also be useful for the development of new bio-

based products. I recommend the publication with minor corrections, according to the attached file.

Line 31, page 1: This sentence needs to be better explained here.

The sentence was rewritten in lines 3-5.

Line 28, page 2: Add location information and geographic coordinates

The information was added in line 33.

Line 29, page 2: Add location information

The information was added in line 35.

Line 9, page 3: Add location

The information was added in line 70.

Line 32, page 3: Explain why you chose this period. Was it based on pre-tests?

The time of 8 days was chosen based on pre-tests. After that period all pupae parasitized by EPNs had died.

Reviewer: 2

Comments to the Author(s)

In the attached file you can see the comments.

Line 6, page 2: It is recommended to add previous studies of the isolation of natural enemies of Eucalyptus pests.

The introduction was improved and new studies on the isolation of natural enemies in eucalyptus pests have been added to the lines 7-15. These studies are related to parasitoids as there are no studies with isolates of nematodes from individuals of *G. platensis*.

Line 16, page 3: Correctly separate the references.

The references were correctly separated.

Line 41, page 3: The first result of the study is the strain of nematodes that were isolated.

The results section was improved.

Line 49, page 3: Describe the results and do not discuss, this must be included in the discussion section.

The results section was rewritten in lines 101-117.

Line 52, page 3: The text should be concise and not repeat materials and methods.

The results were rewritten.

Line 55, page 3: Do not repeat materials and methods.

The results were rewritten.

Line 58, page 3: This determination was not mentioned in the materials and methods section.

The evaluation of the mortality of pupae infected by IJs was mentioned in materials and methods in lines 91-93.

Line 59, page 3: Indicate how it was verified that the pupa was not dead.

Pupae parasitized by EPNS have a darker color with orange or brown tones and no movement. This information was added in text in lines 112.

Line 60, page 3: The description of the results of the concentration of 500 IJs is missing.

This information was added in lines 114-116.

Line 5, page 4: Do not repeat results in the discussion section.

The sentence was rewritten in line 120-123.

Line 15, 21 and 32, page 4: It seems more like a literature review than a discussion.

The discussion section was improved.

Line 25, page 4: Recording 8 days after inoculation the nematodes do not indicate that they died until day 8, most likely they died earlier.

The sentence was rewritten in line 135.

Line 28, page 4: This phrase is not related to the previous text.

The sentence was excluded.

Line 36, page 4: Do not argue things that cannot be evidenced with the results obtained in the present study.

The sentence was excluded.

Line 44, page 4: The terminology used to define virulence is incorrect.

The word 'virulence' was changed for 'pathogenicity'.

Table 2: This evaluation and results were not described in the materials and methods and results section.

The evaluation of emergence of nematodes was added to the materials and methods section on the lines 92-93 and section results in line 112-114.

We hope that all corrections have been addressed as indicated but we are ready to make any other that the reviewers consider necessary.

Sincerely,

Bárbara Monteiro de Castro e Castro